# On the Quantitative Analysis of Decoder-Based Generative Models

**Yuhuai Wu**
Department of Computer Science
University of Toronto
ywu@cs.toronto.edu

**Yuri Burda**
OpenAI
yburda@openai.com

**Ruslan Salakhutdinov**
School of Computer Science
Carnegie Mellon University
rsalakhu@cs.cmu.edu

**Roger Grosse**
Department of Computer Science
University of Toronto
rgrosse@cs.toronto.edu

## Abstract

The past several years have seen remarkable progress in generative models which produce convincing samples of images and other modalities. A shared component of many powerful generative models is a decoder network, a parametric deep neural net that defines a generative distribution. Examples include variational autoencoders, generative adversarial networks, and generative moment matching networks. Unfortunately, it can be difficult to quantify the performance of these models because of the intractability of log-likelihood estimation, and inspecting samples can be misleading. We propose to use Annealed Importance Sampling for evaluating log-likelihoods for decoder-based models and validate its accuracy using bidirectional Monte Carlo. The evaluation code is provided at https://github.com/tonywu95/eval_gen. Using this technique, we analyze the performance of decoder-based models, the effectiveness of existing log-likelihood estimators, the degree of overfitting, and the degree to which these models miss important modes of the data distribution.

## 1 Introduction

In recent years, deep generative models have dramatically pushed forward the state-of-the-art in generative modelling by generating convincing samples of images (Radford et al., 2016), achieving state-of-the-art semi-supervised learning results (Salimans et al., 2016), and enabling automatic image manipulation (Zhu et al., 2016). Many of the most successful approaches are defined in terms of a process which samples latent variables from a simple fixed distribution (such as Gaussian or uniform) and then applies a learned deterministic mapping which we will refer to as a decoder network. Important examples include variational autoencoders (VAEs) (Kingma & Welling, 2014; Rezende et al., 2014), generative adversarial networks (GANs) (Goodfellow et al., 2014), generative moment matching networks (GMMNs) (Li & Swersky, 2015; Dziugaite et al., 2015), and nonlinear independent components estimation (Dinh et al., 2014). We refer to this set of models collectively as decoder-based models, also known as density networks (MacKay & Gibbs, 1998).

While many decoder-based models are able to produce convincing samples (Denton et al., 2015; Radford et al., 2016), rigorous evaluation remains a challenge. Comparing models by inspecting samples is labor-intensive, and potentially misleading (Theis et al., 2016). While alternative quantitative criteria have been proposed (Bounliphone et al., 2016; Im et al., 2016; Salimans et al., 2016), log-likelihood of held-out test data remains one of the most important measures of a generative model's performance. Unfortunately, unless the decoder is designed to be reversible (Dinh et al., 2014; 2016), log-likelihood estimation in decoder-based models is typically intractable. In the case of VAE-based models, a learned encoder network gives a tractable lower bound, but for GANs and GMMNs it is not obvious how even to compute a good lower bound. Even when lower bounds are available, their accuracy may be hard to determine. Because of the difficulty of log-likelihood

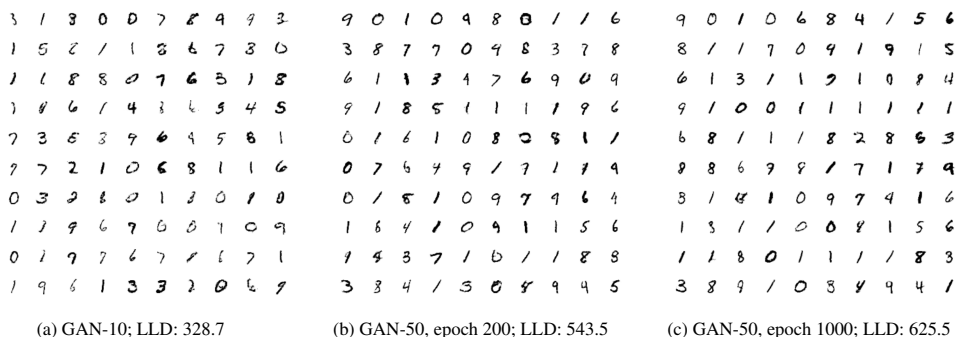

(a) GAN-10; LLD: 328.7          (b) GAN-50, epoch 200; LLD: 543.5          (c) GAN-50, epoch 1000; LLD: 625.5

Figure 1: (a) samples from a GAN with 10 latent dimensions, (b) and (c) samples from a GAN with 50 latent dimensions at different epochs of training. While it is difficult to visually discern differences between these three models, their log-likelihood (LLD) values span almost 300 nats.

evaluation, it is hard to answer basic questions such as whether the networks are simply memorizing training examples, or whether they are missing important modes of the data distribution.

The most widely used estimator of log-likelihood for GANs and GMMNs is the Kernel Density Estimator (KDE) (Parzen, 1962). It estimates the likelihood under an approximation to the model's distribution obtained by simulating from the model and convolving the set of samples with a kernel (typically Gaussian). Unfortunately, KDE is notoriously inaccurate for estimating likelihood in high dimensions, because it is hard to tile a high-dimensional manifold with spherical Gaussians (Theis et al., 2016).

In this paper, we propose to use annealed importance sampling (AIS; (Neal, 2001)) to estimate log-likelihoods of decoder-based generative models and to obtain approximate posterior samples. Importantly, we validate this approach using Bidirectional Monte Carlo (BDMC) (Grosse et al., 2015), which provably bounds the log-likelihood estimation error and the KL divergence from the true posterior distribution for data simulated from a model. For most models we consider, we find that AIS is two orders of magnitude more accurate than KDE, and is accurate enough to perform fine-grained comparisons between generative models. In the case of VAEs, we show that AIS can be further sped up by using the recognition network to determine the initial distribution; this yields an estimator which is fast enough to be run repeatedly during training.

Using the proposed method, we analyze several scientific questions central to understanding decoder-based generative models. First, we measure the accuracy of KDE and of the importance weighting bound which is commonly used to evaluate VAEs. We find that the KDE error is larger than the (quite significant) log-likelihood differences between different models, and that KDE can lead to misleading conclusions. The importance weighted bound, while reasonably accurate, can also yield misleading results in some cases.

Second, we compare the log-likelihoods of VAEs, GANs, and GMMNs, and find that VAEs achieve log-likelihoods several hundred nats higher than the other models (even though KDE considers all three models to have roughly the same log-likelihood). Third, we analyze the degree of overfitting in VAEs, GANs, and GMMNs. Contrary to a commonly proposed hypothesis, we find that GANs and GMMNs are *not* simply memorizing their training data; in fact, their log-likelihood gaps between training and test data are much smaller relative to comparably-sized VAEs. Finally, by visualizing (approximate) posterior samples obtained from AIS, we observe that GANs miss important modes of the data distribution, even ones which are represented in the training data.

We emphasize that none of the above phenomena can be measured using KDE or the importance weighted bound, or by inspecting samples. (See Fig. 1 for an example where it is tricky to compare models based on samples.) While log-likelihood is by no means a perfect measure, we find that the ability to accurately estimate log-likelihoods of decoder-based models yields crucial insight into their behavior and suggests directions for improving them.

## 2 BACKGROUND

### 2.1 DECODER-BASED GENERATIVE MODELS

In generative modelling, a decoder network is often used to define a generative distribution by transforming samples from some simple distribution (e.g. normal) to the data manifold. In this

paper, we consider three kinds of decoder-based generative models: Variational Autoencoder (VAE) (Kingma & Welling, 2014), Generative Adversarial Network (GAN) (Goodfellow et al., 2014), and Generative Moment Matching Network (GMMN) (Li & Swersky, 2015; Dziugaite et al., 2015).

### 2.1.1 VARIATIONAL AUTOENCODER

A variational autoencoder (VAE) (Kingma & Welling, 2014) is a probabilistic directed graphical model. It is defined by a joint distribution over a set of latent random variables $z$ and the observed variables $x$: $p(x, z) = p(x|z)p(z)$. The prior over the latent random variables, $p(z)$, is usually chosen to be a standard Gaussian distribution. The data likelihood $p(x|z)$ is usually a Gaussian or Bernoulli distribution whose parameters depend on $z$ through a deep neural network, known as the decoder network. It also uses an approximate inference model called an encoder or recognition network, that serves as a variational approximation $q(z|x)$ to the posterior $p(z|x)$. The decoder network and the encoder networks are jointly trained to maximize the evidence lower bound (ELBO):

$$\log p(x) \geq \mathbb{E}_{q(z|x)}[\log p(x|z)] - KL(q(z|x)||p(z)) \tag{1}$$

In addition, the reparametrization trick is used to reduce the variance of the gradient estimate.

### 2.1.2 GENERATIVE ADVERSARIAL NETWORK (GAN)

A generative adversarial network (GAN) (Goodfellow et al., 2014) is a generative model trained by a game between a decoder network and a discriminator network. It defines the generative model by sampling the latent variable $z$ from some simple prior distribution $p(z)$ (e.g., Gaussian) followed through the decoder network. The discriminator network $D(\cdot)$ outputs a probability of a given sample coming from the data distribution. Its task is to distinguish samples from the generator distribution from real data. The decoder network, on the other hand, tries to produce samples as realistic as possible, in order to fool the discriminator into accepting its outputs as being real. The competition between the two networks results in the following minimax problem:

$$\min_G \max_D \mathbb{E}_{x \sim p_{data}}[\log D(x)] + \mathbb{E}_{z \sim p(z)}[\log(1 - D(G(z)))] \tag{2}$$

Unlike VAE, the objective is not explicitly related to the log-likelihood of the data. Moreover, the generative distribution is a deterministic mapping, i.e., $p(x|z)$ is a Dirac delta distribution, parametrized by the deterministic decoder. This can make data likelihood ill-defined, as the probability density of any particular point $x$ can be either infinite, or exactly zero.

### 2.1.3 GENERATIVE MOMENT MATCHING NETWORK (GMMN)

Generative moment matching networks (GMMNs) (Li & Swersky, 2015; Dziugaite et al., 2015) adopt maximum mean discrepancy (MMD) as the training objective, a moment matching criterion where kernel mean embedding techniques are used to avoid unnecessary assumptions of the distributions. It has the same issue as GAN in that the log-likelihood is undefined.

## 2.2 ANNEALED IMPORTANCE SAMPLING

We are interested in estimating the probability $p(x) = \int p(z)p(x|z)\,dz$ a model assigns to an observation $x$. This is equivalent to computing the normalizing constant of the unnormalized distribution $f(z) = p(z, x)$. One naïve approach is likelihood weighting, where one samples $\{z^{(k)}\}_{k=1}^K \sim p(z)$ and averages the conditional likelihoods $p(x|z^{(k)})$. This is justified by the following identity:

$$p(x) = \int \frac{p(x, z)}{p(z)}p(z)\,dz = \mathbb{E}_{z \sim p(z)}[p(x|z)] \tag{3}$$

Likelihood weighting can be viewed as simple importance sampling, where the proposal distribution is the prior $p(z)$ and the target distribution is the posterior $p(z|x)$. Unfortunately, importance sampling works well only when the proposal distribution is a good match for the target distribution. For the models considered in this paper, the (very broad) prior can be drastically different than the (highly concentrated) posterior, leading to inaccurate estimates of the likelihood.

Annealed importance sampling (AIS; Neal, 2001) is a Monte Carlo algorithm commonly used to estimate (ratios of) normalizing constants. Roughly speaking, it computes a sequence of importance

sampling based estimates, each of which is stable because it involves two distributions which are very similar. In particular, suppose one is interested in estimating the normalizing constant $\mathcal{Z} = \int f(z) \, \mathrm{d}z$ of an unnormalized distribution $f(z)$. (In the likelihood estimation setting, $f(z) = p(z, x)$ and $\mathcal{Z} = p(x)$.) One must specify a sequence of distributions $q_1, ..., q_T$, where $q_t = f_t / Z_t$, and $f_T = f$ is the target distribution. It is required that one can obtain one or more exact samples from the initial distribution $q_1$. One must also specify a sequence of reversible MCMC transition operators $\mathcal{T}_1, ..., \mathcal{T}_T$, where $\mathcal{T}_t$ leaves $q_t$ invariant.

AIS produces a (nonnegative) unbiased estimate of the ratio $\mathcal{Z}_T / \mathcal{Z}_1$ as follows: first, we sample a random initial state $z_1 \sim q_1$ and set the initial weight $w_1 = 1$. For every stage $t \geq 2$ we update the weight $w$ and sample the state $z_t$ according to

$$w_t \leftarrow w_{t-1} \frac{f_t(z_{t-1})}{f_{t-1}(z_{t-1})} \qquad z_t \sim \mathcal{T}_t(z|z_{t-1}) \tag{4}$$

As demonstrated by Neal (2001), this procedure produces a nonnegative weight $w_T$ such that $\mathbb{E}[w_T] = \mathcal{Z}_T / \mathcal{Z}_1$. Typically, $\mathcal{Z}_1$ is known, so one computes multiple independent AIS weights $\{w_T^{(K)}\}_{k=1}^K$ and obtains the unbiased estimate $\hat{\mathcal{Z}}_T = \mathcal{Z}_1 \frac{1}{K} \sum_{k=1}^K w_T^{(K)}$. In the likelihood estimation setting, $\mathcal{Z}_1 = 1$ and $\mathcal{Z}_T = p(x)$, so we denote this estimator as $\hat{p}(x)$.

Typically, the unnormalized intermediate distributions are simply defined to be geometric averages $f_t(z) = f_1(z)^{1-\beta_t} f_T(z)^{\beta_t}$, where the $\beta_t$ are monotonically increasing parameters with $\beta_1 = 0$ and $\beta_T = 1$. For $f_1(z) = p(z)$ and $f_T(z) = p(z, x)$, this gives

$$f_t(z) = p(z) \, p(x|z)^{\beta_t}. \tag{5}$$

As shown by Neal (2001), under certain regularity conditions, the variance of $\hat{\mathcal{Z}}_T$ tends to zero as the number of intermediate distributions is increased. AIS is very effective in practice, and has been used to estimate normalizing constants of complex high-dimensional distributions (Salakhutdinov & Murray, 2008).

## 2.3 BIDIRECTIONAL MONTE CARLO

AIS provides a nonnegative unbiased estimate $\hat{p}(x)$ of $p(x)$. However, it is often more practical to estimate $p(x)$ in the log space, i.e. $\log p(x)$, because of underflow problem of dealing with many products of probability measure. In general, we note that logarithm of a nonnegative unbiased estimate is a *stochastic lower bound* of the log estimand (Grosse et al., 2015). In particular, $\log \hat{p}(x)$ is a stochastic lower bound on $\log p(x)$, satisfying $\mathbb{E}[\log \hat{p}(x)] \leq \log p(x)$ and $\Pr(\log \hat{p}(x) > \log p(x) + b) < e^{-b}$.

Grosse et al. (2015) pointed out that if AIS is run in reverse starting from an exact posterior sample, it yields an unbiased estimate of $1/p(x)$, which (by the above argument) can be seen as a stochastic *upper* bound on $\log p(x)$. The combination of lower and upper bounds from forward and reverse AIS is known as bidirectional Monte Carlo (BDMC). In many cases, the combination of bounds can pinpoint the true value quite precisely. While posterior sampling is just as hard as log-likelihood estimation (Jerrum et al., 1986), in the case of log-likelihood estimation for *simulated* data, one has available a single exact posterior sample: the parameters and/or latent variables which generated the data. Because this trick is only applicable to simulated data, BDMC is most useful for measuring the accuracy of a log-likelihood estimator on simulated data.

Grosse et al. (2016) observed that BDMC can also be used to validate posterior inference algorithms, as the gap between upper and lower bounds is itself a bound on the KL divergence of approximate samples from the true posterior distribution.

## 3 METHODOLOGY

For a given generative distribution $p(x, z) = p(z)p(x|z)$, our task is to measure the log-likelihood of test examples $\log p(x_{test})$. We first discuss how we define the generative distribution for decoder-based networks. For VAE, the generative distribution is defined in the standard way, where $p(z)$ is a standard normal distribution and $p(x|z)$ is a normal distribution parametrized by mean $\mu_\theta(z)$ and $\sigma_\theta(z)$, predicted by the generator given the latent code. However, the observation distribution for GANs and GMMNs is typically taken to be a delta function, so that the model's distribution covers

only a submanifold of the space of observables. In order for the likelihood to be well-defined, we follow the same assumption made when evaluating using Kernel Density Estimator (Parzen, 1962): we assume a Gaussian observation model with a fixed variance hyperparameter $\sigma^2$. We will refer to the distribution defined by this Gaussian observation model as $p_\sigma$.

Observe that the KDE estimate is given by

$$\hat{p}_\sigma(x) = \frac{1}{K} \sum_{k=1}^{K} p_\sigma(x|z^{(k)}), \tag{6}$$

where $\{z^{(k)}\}_{k=1}^{K}$ are samples from the prior $p(z)$. This is equivalent to likelihood weighting for the distribution $p_\sigma$, which is an instance of simple importance sampling (SIS). Because SIS is an unbiased estimator of the likelihood, $\log \hat{p}_\sigma(x)$ is a stochastic lower bound on $\log p_\sigma(x)$ (Grosse et al., 2015). Unfortunately, SIS can result in very poor estimates when the evidence has low prior probability (i.e. the posterior is very dissimilar to the prior). This suggests that AIS might be able to yield much more accurate log-likelihood estimates under $p_\sigma$. We note that KDE can be viewed as a special case of AIS where the number of intermediate distributions is set to 0.

We now describe specifically how we carry out evaluation using AIS. In most of our experiments, we choose the initial distribution of AIS to be $p(z)$, the same prior distribution used in training decoder-based models. If the model provides an encoder network (e.g., VAE), we can take the approximated distribution predicted by the encoder $q(z|x)$ as the initial distribution of the AIS chain. For continuous data, we define the unnormalized density of target distribution to be the joint generative distribution with the Gaussian noise model, $p_\sigma(x, z) = p_\sigma(x|z)p(z)$. For the small subset of experiments done on the binary data, we define the observation model to be a Bernoulli model with mean predicted by the decoder. Our intermediate distributions are geometric averages of the prior and posterior, as in Eqn. 5. Since all of our experiments are done using continuous latent space, we use Hamiltonian Monte Carlo (Neal, 2010) as the transition operator for sampling latent samples along annealing. The evaluation code is provided at `https://github.com/tonywu95/eval_gen`.

## 4 RELATED WORK

AIS is known to be a powerful technique of estimating the partition function of the model. One influential example was the use of AIS to evaluate deep belief networks (Salakhutdinov & Murray, 2008). Although we used the same technique, the problem we consider is completely different. First of all, the model they consider is undirected graphical models, whereas decoder-based models are directed graphical models. Secondly, their model has a well-defined probabilistic density function in terms of energy function, whereas we need to consider different probabilistic model for one in which the the likelihood is ill-defined. In addition, we validate our estimates using BDMC.

Theis et al. (2016) give an in-depth analysis of issues that might come up in evaluating generative models. They also point out that a model that completely fails at modelling the proportion of modes of the distribution might still achieve a high likelihood score. Salimans et al. (2016) propose an image-quality measure which they find to be highly correlated with human visual judgement. They propose to feed the samples $x$ of the model to the "inception" model to obtain a conditional label distribution $p(y|x)$, and evaluate the score defined by $\exp \mathbb{E}_x \text{KL}(p(y|x)||p(y))$, which is motivated by having a low entropy of $p(y|x)$ but a large entropy of $p(y)$. However, the measure is largely based on visual quality of the sample, and we argue that the visual quality can be a misleading way to evaluate a model.

## 5 EXPERIMENTS

### 5.1 DATASETS

All of our experiments were performed on the MNIST dataset of images of handwritten digits (LeCun et al., 1998). For consistency with prior work on evaluating decoder-based models, most of our experiments used the continuous inputs. We dequantized the data following Uria et al. (2013), by adding a uniform noise of $\frac{1}{256}$ to the data and rescaling it to be in $[0, 1]^D$ after dequantization. We use the standard split of MNIST into 60,000 training and 10,000 test examples, and used 50,000 images from the training set for training, and remaining 10,000 images for validation. In addition,

some of our experiments used the binarized MNIST dataset with a Bernoulli observation model (Salakhutdinov & Murray, 2008).

## 5.2 MODELS

For most of our experiments, we considered two decoder architectures: a small one with 10 latent dimensions, and a larger one with 50 latent dimensions. We use standard Normal distribution as prior for training all of our models. All layers were fully connected, and the number of units in each layer was 10–64–256–256-1024–784 for the smaller architecture and 50–1024–1024–1024–784 for the larger one. We trained both architectures using the VAE, GAN, and GMMN objectives, resulting in six networks which we refer to as VAE-10, VAE-50, etc. In general, the larger architecture performed substantially better on both the training and test sets, but we analyze the smaller architecture as well because it better highlights some of the differences between the training criteria. Additional architectural details are given in Appendix A.1.

In order to enable a direct comparison between training criteria, all models used a spherical Gaussian observation model with fixed variance. This is consistent with previous protocols for evaluating GANs and GMMNs. However, we note that this observation model is a nontrivial constraint on the VAEs, which could instead be trained with a more flexible diagonal Gaussian observation model where the variances depend on the latent state. Such observation models can easily achieve much higher log-likelihood scores, for instance by noticing that boundary pixels are always close to 0. (E.g., we trained a VAE with the more general observation model which achieved a log-likelihood of at least 2200 nats on continuous MNIST.) Therefore, the log-likelihood values we report should not be compared directly against networks which have a more flexible observation model.

## 5.3 VALIDATION OF LOG-LIKELIHOOD ESTIMATES

Before we analyze the performance of the trained networks, we must first determine the accuracy of the log-likelihood estimators. In this section, we validate the accuracy of our AIS-based estimates using BDMC. We then analyze the error in the KDE and IWAE estimates and highlight some cases where these measures miss important phenomena.

### 5.3.1 VALIDATION OF AIS

We used AIS to estimate log-likelihoods for all models under consideration. Except where otherwise specified, all AIS estimates were obtained using 16 independent chains, 10,000 intermediate distributions of the form in Eqn. 5, and a transition operator consisting of one proposed HMC trajectory with 10 leapfrog steps.[1] Following Ranzato et al. (2010), the HMC stepsize was tuned to achieve an acceptance rate of 0.65 (as recommended by Neal (2010)).

For all six models, we evaluated the accuracy of this estimation procedure using BDMC on data sampled from the model's distribution on 1000 simulated examples. The gap between the log-likelihood estimates produced by forward AIS (which gives a lower bound) and reverse AIS (which gives an upper bound) bounds the error of the AIS estimates on simulated data. We refer to this gap as the *BDMC gap*. For five of the six networks under consideration, we found the BDMC gap to be less than 1 nat. For the remaining model (GAN-50), the gap was about 10 nats. Both gaps are much smaller than our measured log-likelihood differences between models. If these gaps are representative of the true error in the estimates on the real data, then this indicates AIS is accurate enough to make fine-grained comparisons between models and to benchmark other log-likelihood estimators. (The BDMC gap is not guaranteed to hold for the real data, although Grosse et al. (2016) found the behavior of AIS to match closely between real and simulated data.)

### 5.3.2 HOW ACCURATE IS KERNEL DENSITY ESTIMATION?

Kernel density estimation (KDE) (Parzen, 1962) is widely used to evaluate decoder-based models (Goodfellow et al., 2014; Li & Swersky, 2015), and a variant was proposed in the setting of evaluating Boltzmann machines (Bengio et al., 2013). Papers reporting KDE estimates often caution that the

---

[1]We used the HMC implementation from `http://deeplearning.net/tutorial/deeplearning.pdf`

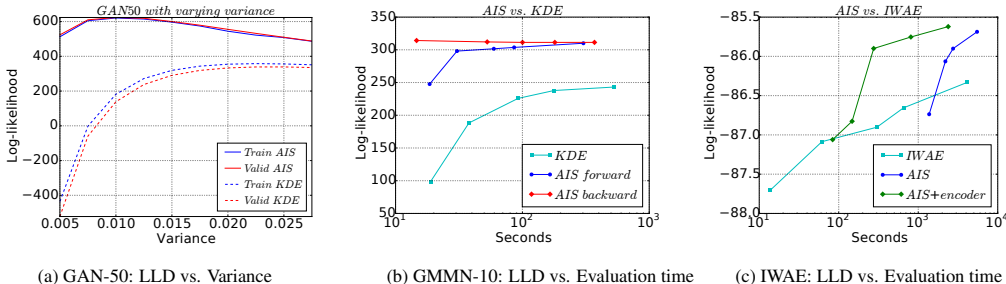

(a) GAN-50: LLD vs. Variance (b) GMMN-10: LLD vs. Evaluation time (c) IWAE: LLD vs. Evaluation time

Figure 2: (a) Log-likelihood of GAN-50, under different choices of variance parameter. (b) Log-likelihood of GMMN-10 on 100 simulated examples evaluated by AIS and KDE vs. the corresponding running time. We show the BDMC gap converges to almost zero as we increase the running time. (c) Log-likelihood of IWAE on 10,000 test examples evaluated by AIS and IWAE bound vs. running time. (a), (b) are results on continuous MNIST, and (c) is on binarized MNIST. Note that AIS/AIS+encoder dominates the other estimate in both estimation accuracy and running time.

| (Nats) | AIS | AIS+encoder | IWAE bound | # dist AIS | # dist AIS+encoder | # samples |
|---|---|---|---|---|---|---|
| IWAE | -85.679 | -85.754 | -86.902 | 1000 | 100 | 10000 |
| | -85.619 | -85.621 | -86.464 | 10000 | 1000 | 100000 |

Table 1: AIS vs. IWAE bound on 10,000 test examples of binarized MNIST. "# dist" denotes the number of intermediate distributions used for evalution. We find AIS estimate is consistently 1 nat higher than IWAE bound; AIS+encoder can achieve about the same estimate as AIS, but with 1 order of magnitude less number of intermediate distributions.

KDE is not meant to be applied in high-dimensional spaces and that the results might therefore be inaccurate. Nevertheless, KDE remains the standard protocol for evaluating decoder-based models. We analyzed the accuracy of the KDE estimates by comparing against AIS. Both estimates are stochastic lower bounds on the true log-likelihood (see Section 3), so larger values are guaranteed (with high probability) to be more accurate.

For each estimator, we varied one parameter influencing the computational budget; for AIS, this was the number of intermediate distributions (chosen from $\{100, 500, 1000, 2000, 10000\}$), and for KDE, it was the number of samples (chosen from $\{10000, 100000, 500000, 1000000, 2000000\}$). Using GMMN-10 for illustration, we plot both log-likelihood estimates 100 simulated examples as a function of evaluation time in Fig. 2(b). We also plot the upper bound of likelihood given by running AIS in reverse direction. We see that the BDMC gap approaches to zero, validating the accuracy of AIS. We also see that the AIS estimator achieves much more accurate estimates during similar evaluation time. Furthermore, the KDE estimates appear to level off, suggesting one cannot obtain accurate results even using orders of magnitude more samples.

The KDE estimation error also impacts the estimate of the observation noise $\sigma$, since a large value of $\sigma$ is needed for the samples to cover the full distribution. We compared the log-likelihoods estimated by AIS and KDE with varying choices of $\sigma$ on 100 training and validation examples of MNIST. We used 1 million simulated samples for KDE evaluation, which takes almost the same time as running AIS estimation. In Fig. 2(a), we show the log-likelihood of GAN-50 estimated by KDE and AIS as a function of $\sigma$. Because the accuracy of KDE declines sharply for small $\sigma$ values, it creates a strong bias towards large $\sigma$.

### 5.3.3 HOW ACCURATE IS THE IWAE BOUND?

In principle, one could estimate VAE likelihoods using the VAE objective function (which is a lower bound on the true log-likelihood). However, it is more common to use importance weighting, where the proposal distribution is computed by the recognition network. This is provably more accurate than the VAE bound (Burda et al., 2016). Because the importance weighted estimate corresponds to the objective function used by the Importance Weighted Autoencoder (IWAE) (Burda et al., 2016), we will refer to it as the *IWAE bound*.

On continuous MNIST, the IWAE bound underestimated the true log-likelihoods by at least 33.2 nats on the training set and 187.4 nats on the test set. While this is considerably more accurate than KDE, the error is still significant. Interestingly, this result also suggests that the recognition network overfits the training data.

| (Nats) | AIS Test | AIS Train | BDMC gap | KDE Test | IWAE Test |
|---|---|---|---|---|---|
| VAE-50 | 991.435±6.477 | 1298.830±0.863 | 1.540 | 351.213 | 826.325 |
| GAN-50 | 627.297±8.813 | 648.283±21.115 | 10.045 | 300.331 | / |
| GMMN-50 | 593.472±8.591 | 607.272±1.451 | 1.146 | 277.193 | / |
| VAE-10 | 705.375±7.411 | 791.029±0.810 | 0.832 | 408.659 | 486.466 |
| GAN-10 | 328.772±5.538 | 346.640±4.260 | 0.934 | 259.673 | / |
| GMMN-10 | 346.679±5.860 | 358.943±6.485 | 0.605 | 262.73 | / |

Table 2: Model comparisons on 1000 test and training examples of continuous MNIST. Confidence intervals reflect the variability from the choice of training or test examples (which appears to be the dominant source of error for the AIS values). AIS, KDE, and IWAE are all stochastic lower bounds on the log-likelihood.

Since VAE and IWAE results have customarily been reported on binarized MNIST, we additionally trained an IWAE in this setting. The training details are given in Appendix A.2. To show the practicality of our method, we evaluated the IWAE on the full 10000 test using AIS and IWAE bound, with different choices of intermediate distribution and number of simulated samples, shown in Table 1. We also evaluate AIS with the initial distribution defined by encoders of VAEs, denoted as AIS+encoder. We find that the IWAE bound underestimates the true value by at least 1 nat, which is a large difference by the standards of binarized MNIST. (E.g., it represents about half of the gap between a state-of-the-art permutation-invariant model (Tran et al., 2016) and one which exploits structure (van den Oord et al., 2016).) The AIS and IWAE estimates are compared in terms of evaluation time in Fig. 2 (c).

## 5.4 SCIENTIFIC FINDINGS

Having validated the accuracy of AIS, we now use it to analyze the effectiveness of various training criteria. We also highlight phenomena which would not be observable using existing log-likelihood estimators or by inspecting samples. For all experiments in this section, we used 10,000 intermediate distributions for AIS, 1 million simulated samples for KDE, and 200,000 importance samples for the IWAE bound. (These settings resulted in similar computation time for all three estimators.)

### 5.4.1 MODEL LIKELIHOOD COMPARISON

We evaluated the trained models using AIS and KDE on 1000 test examples of MNIST; results are shown in Table 2. We find that for all three training criteria, the larger architectures consistently outperformed the smaller ones. We also find that for both the 10- and 50-dimensional architectures, the VAEs achieved substantially higher log-likelihoods than GANs or GMMNs. It is not surprising that the VAEs achieved higher likelihood, because they were trained using a likelihood-based objective while the GANs and GMMNs were not. However, it is interesting that the difference in log-likelihoods was so large; in the rest of this section, we attempt to analyze what exactly is causing this large difference.

We note that the KDE errors were of the same order of magnitude as the differences between models, indicating that it cannot be used reliably to compare log-likelihoods. Furthermore, KDE did not identify the correct ordering of models; for instance, it estimated a lower log-likelihood for VAE-50 than for VAE-10, even though its true log-likelihood was almost 300 nats higher. KDE also underestimated by an order of magnitude the log-likelihood improvements that resulted from using the larger architectures. (E.g., it estimated a 15 nat difference between GMMN-10 and GMMN-50, even though the true difference was 247 nats as estimated by AIS.)

These differences are also hard to observe simply by looking at samples; for instance, we were unable to visually distinguish the quality of samples for GAN-10 and GAN-50 (see Fig. 1), even though their log-likelihoods differed by almost 300 nats on both the training and test sets.

### 5.4.2 MEASURING THE DEGREE OF OVERFITTING

One question that arises in evaluation of decoder-based generative models is whether they memorize parts of the training dataset. One cannot test this by looking only at model samples. The commonly reported nearest-neighbors from the training set can be misleading (Theis et al., 2016), and interpolation in the latent space between different samples can be visually appealing, but does not provide a quantitative measure of the degree of generalization.

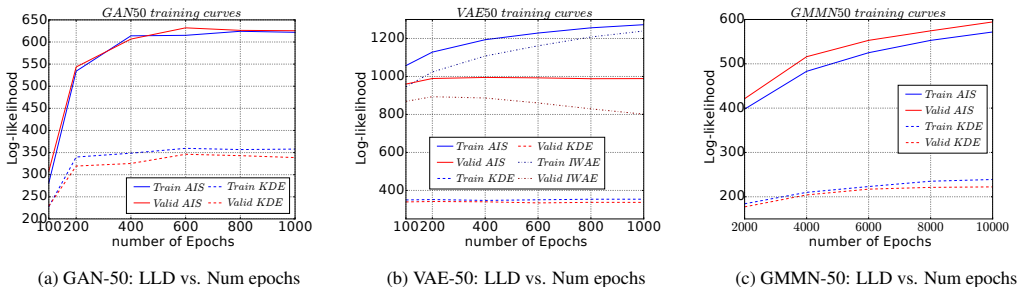

(a) GAN-50: LLD vs. Num epochs    (b) VAE-50: LLD vs. Num epochs    (c) GMMN-50: LLD vs. Num epochs

Figure 3: Training curves for (a) GAN-50, (b) VAE-50, and (c) GMMN-10, as measured by AIS, KDE, and (if applicable) the IWAE lower bound. All estimates shown here are lower bounds. In (c), the gap between training and validation log-likelihoods is not fairly small (see Table 2).

To analyze the degree of overfitting, Fig. 3 shows training curves for three networks as measured by AIS, KDE, and the IWAE bound. We observe that GAN-50's training and test log-likelihoods are nearly identical throughout training, disconfirming the hypothesis that it was memorizing training data. Both GAN-50 and GMMN-50 overfit less than VAE-50.

We also observed two phenomena which could not be measured using existing techniques. First, in the case of VAE-50, the IWAE lower bound starts to decline after 200 epochs, while the AIS estimates hold steady, suggesting it is the recognition network rather than the generative network which is overfitting most. Second, the GMMN-50 training and validation error continue to improve at 10,000 epochs, even though KDE erroneously indicates that performance has leveled off.

### 5.4.3    How appropriate is the observation model?

Appendix B addresses the questions of whether the spherical Gaussian observation model is a good fit and whether the log-likelihood differences could be an artifact of the observation model. We find that all of the models can be substantially improved by accounting for non-Gaussianity, but that this effect is insufficient to explain the gap between the VAEs and the other models.

### 5.4.4    Are the networks missing modes?

It was previously observed that one of the potential failure modes of Boltzmann machines is to fail to generate one or more modes of a distribution or to drastically misallocate probability mass between modes (Salakhutdinov & Murray, 2008). Here we analyze this for decoder-based models.

First, we ask a coarse-grained version of this question: do the networks allocate probability mass correctly between the 10 digit classes, and if not, can this explain the difference in log-likelihood scores? In Fig. 1, we see that GAN-50's distribution of digit classes was heavily skewed: out of 100 samples, it generated 37 images of 1's, but only a single 2. This appears to be a large effect, but it does not explain the magnitude of the log-likelihood difference from VAEs. In particular, if the allocation of digit classes were off by a factor of 10, this effect by itself could cost at most $\log 10 \approx 2.3$ nats of log-likelihood. Since VAE-50 outperformed GAN-50 by 364 nats, this effect cannot explain the difference.

However, MNIST has many factors of variability beyond simply the 10 digit classes. In order to determine whether any of the models missed more fine-grained modes, we visualized posterior samples for each model conditioned on training and test images. In particular, for each image $x$ under consideration, we used AIS to approximately sample $z$ from the posterior distribution $p(z|x)$, and then ran the decoder on $z$. While these samples are approximate, Grosse et al. (2016) point out that the BDMC gap also bounds the KL divergence of approximate samples from the true posterior. With the exception of GAN-50, our BDMC gaps were on the order of 1 nat, suggesting our approximate posterior samples are fairly representative. The results are shown in Fig. 4. Further posterior visualizations for digit class 2 (the most difficult for the models we considered) are shown in Appendix C.

Both VAEs' posterior samples match the observations almost perfectly. (We observed a few poorly reconstructed examples on the test set, but not on the training set.) The GANs and GMMNs fail to

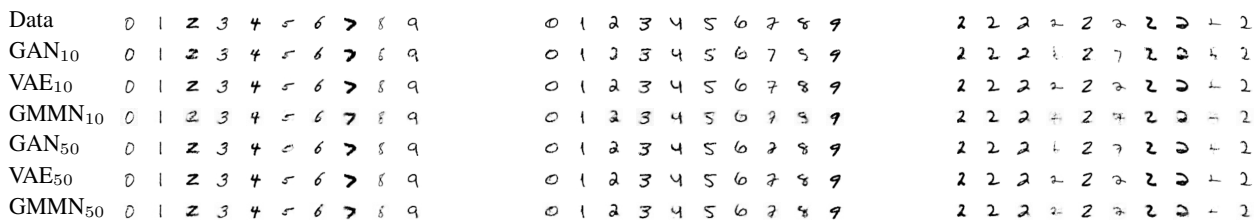

Data
GAN$_{10}$
VAE$_{10}$
GMMN$_{10}$
GAN$_{50}$
VAE$_{50}$
GMMN$_{50}$

(a) The visualization of posterior of 10 training examples

(b)The visualization of posterior of 10 validation examples

(c)The visualization of posterior of 10 examples of digit "2" of training set

Figure 4: (a) and (b) show visualization of posterior samples of 10 training/validation examples. (c) shows visualization of posterior samples of 10 training examples of digit "2". Each column of 10 digits comes from true data and the six models. The order of visualization is: True data, GAN-10, VAE-10, GMMN-10, GAN-50, VAE-50, GMMN-50.

reconstruct some of the examples on both the training and validation sets, suggesting that they failed to learn some modes of the distribution.

ACKNOWLEDGMENTS

We like to thank Yujia Li for providing his original GMMN model and codebase, and thank Jimmy Ba for advice on training GANs. Ruslan Salakhutdinov is supported in part by Disney and ONR grant N000141310721. We also thank the developers of Lasagne (Battenberg et al., 2014) and Theano (Al-Rfou et al., 2016).

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

## A  NETWORK ARCHITECTURES/TRAINING

### A.1  MODELS ON CONTINUOUS MNIST

The decoders have all fully connected layers, and the number of units in each layer was 10–64–256–256-1024–784 for the smaller architecture and 50–1024–1024–1024–784 for the larger one. Other architecture details are summarized as follows.

- For GAN-10, we used a discriminator with the architecture 784-512-256-1, where each layer used dropout with parameter 0.5. For GAN-50, we used a discriminator with architecture 784-4096-4096-4096-4096-1. All hidden layers used dropout with parameter 0.8. All hidden layers in both networks used the tanh activation function, and the output layers used the logistic function.
- The larger model uses an encoder of an architecture 784-1024-1024-1024-100. We add dropout layer between each hidden layer, with a dropout rate of 0.2. The smaller model uses an encoder of an architecture 784-256-64-20. Generator's hidden layers use tanh activation function, and the output layer uses sigmoid unit. Encoder's hidden layers use tanh activation function, and the output layer uses linear activation.
- GMMN: The hidden layers use ReLU activation function, and the output layer uses sigomid unit.

For training GAN/VAE, we use our own implementation. We use Adam for optimization, and perform grid search of learning rate from $\{0.001, 0.0001, 0.00001\}$. For training GMMN, we take the implementation from `https://github.com/yujiali/gmmn.git`. Following the implementation, we use SGD with momentum for optimization, and perform grid search of learning rate from $\{0.1, 0.5, 1, 2\}$, with momentum 0.9.

### A.2  MODELS ON BINARIZED MNIST

Its decoder has the architecture 50-200-200-784 with all tanh hidden layers and sigmoid output layer, and its encoder is symmetric in architecture, with linear output layer. We take the implementation at `https://github.com/yburda/iwae.git` for training the IWAE model.The IWAE bound was computed with 50 samples during training. We keep all the hyperparameter choices the same as in the implementation.

## B  HOW PROBLEMATIC IS THE GAUSSIAN OBSERVATION MODEL?

| (Nats) | Train | | | Valid | | |
|---|---|---|---|---|---|---|
| | Optimal | Fixed | Improvement | Optimal | Fixed | Improvement |
| GAN-50 | 711.405 | 620.498 | 90.907 | 702.699 | 623.492 | 79.207 |
| GMMN-50 | 655.807 | 571.803 | 84.004 | 661.652 | 594.612 | 67.040 |
| GAN-10 | 376.788 | 318.948 | 57.840 | 368.585 | 316.614 | 51.971 |
| GMMN-10 | 393.976 | 345.177 | 48.799 | 371.325 | 332.360 | 38.965 |

Table 3: Optimal variance vs. Fixed variance

In this section, we consider whether the difference in log-likelihood between models could be an artifact of the Gaussian noise model (which we know to be a poor fit). In principle, the Gaussian noise assumption could be unfair to the GANs and GMMNs, because the VAE training uses the correct

observation model, while the GAN and GMMN objectives do not have any particular observation model built in.

To determine the size of this effect, we evaluated the models under a different regime where, instead of choosing a fixed value of the observation noise $\sigma$ on a validation set, $\sigma$ was tuned independently for *each* example.[2] This is not a proper generative model, but it can be viewed as an *upper bound* on the log-likelihood that would be achievable with a heavy-tailed and radially symmetric noise model.[3] Results are shown in Table 3. We see that adapting $\sigma$ for each example results in a log-likelihood improvement between 30 and 100 nats for all of the networks. In general, the examples which show the largest performance jump are images of 1's (which prefer smaller $\sigma$) and 2's (which prefer larger $\sigma$). This is a significant effect, and suggests that one could significantly improve the log-likelihood scores by picking a better observation model. However, this effect is smaller in magnitude than the differences between VAE and GAN/GMMN log-likelihoods, so it fails to explain the likelihood difference.

## C    POSTERIOR VISUALIZATION OF DIGIT "2"

According to the log-likelihood evaluation, we find digit "2" is the hardest digit for modelling. In this section we investigate the quality of modelling "2" of each model. We randomly sampled a fixed set of 100 samples of digit "2" from training data and compare whether model capture this mode. We show the plots of "2" for GAN-10, GAN-50, VAE-10 and true data in the following figures for illustration. We see that GAN-10 fails at capturing many instances of digit "2" in the training data! We see instead of generating "2", it tries to generate digit "1", "7" "9", "4", "8" from reconstruction. GAN-50 does much better, its reconstruction are all digit "2" and there is only some style difference from the true data. VAE-10 totally dominates this competition, where it perfectly reconstructs all the samples of digit "2". We emphasize if directly sampling from each model, samples look visually indistinguishable (see Fig. 1), but we can clearly see differences in posterior samples.

---

[2]We pick the best variance parameter among $\{0.005, 0.01, 0.015, 0.02, 0.025\}$ for each training/validation examples when evaluating GAN-50 and GMMN-50 and $\{0.015, 0.02, 0.025, 0.03, 0.035\}$ when evaluating GAN-10 and GMMN-10.

[3]In particular, heavy-tailed radially symmetric distributions can be viewed as Gaussian scale mixtures (Wainwright & Simoncelli, 1999). I.e., one has a prior distribution on $\sigma$ (possibly learned) and integrates it out for each test example. Clearly the probability under such a mixture cannot exceed the maximum value with respect to $\sigma$.

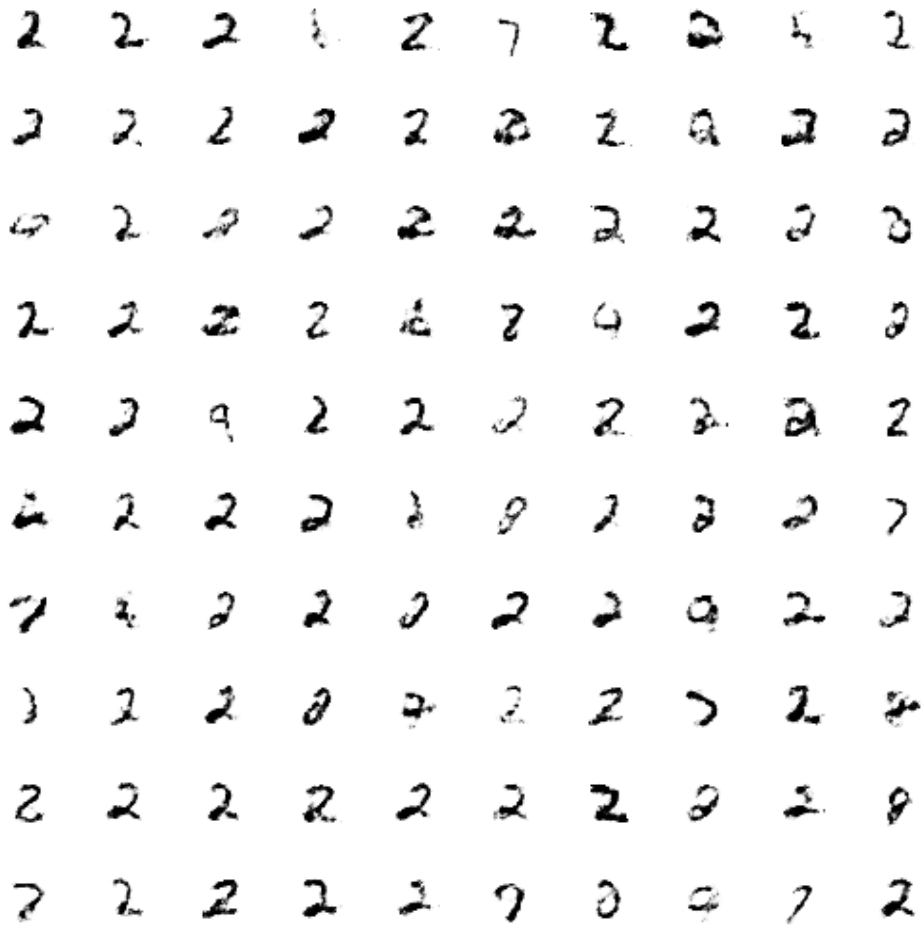

Figure 5: Posterior samples of digit "2" for GAN-10.

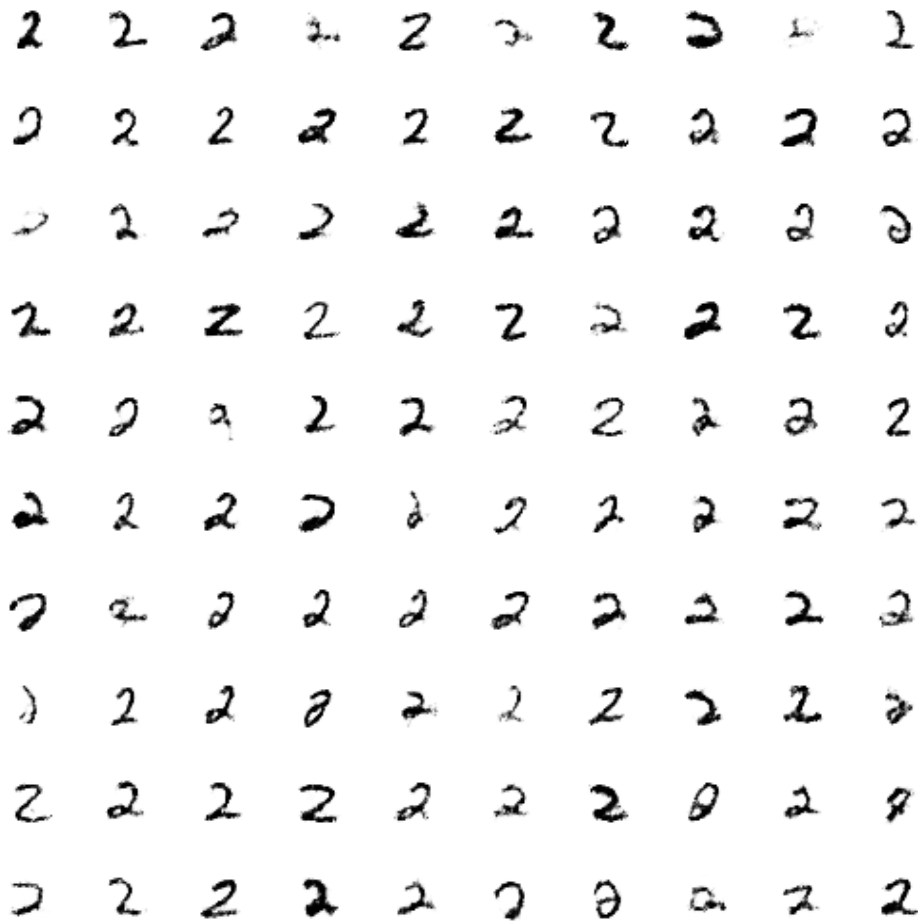

Figure 6: Posterior samples of digit "2" for GAN-50.

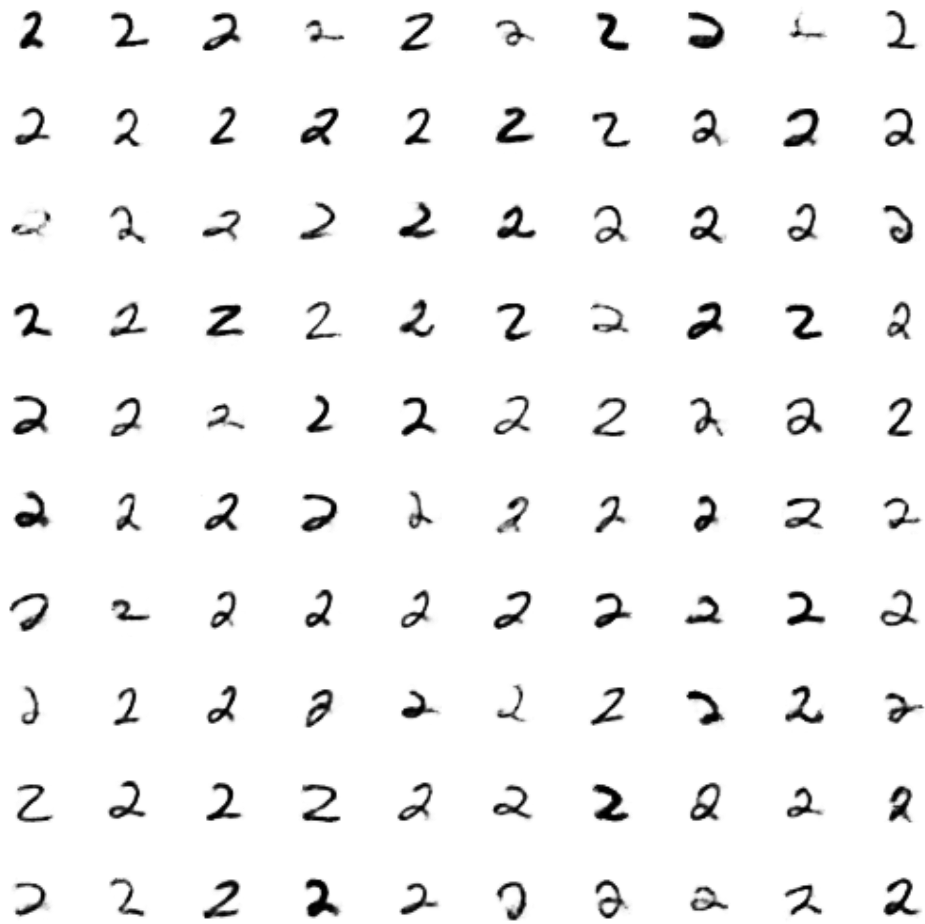

Figure 7: Posterior samples of digit "2" for VAE-10.

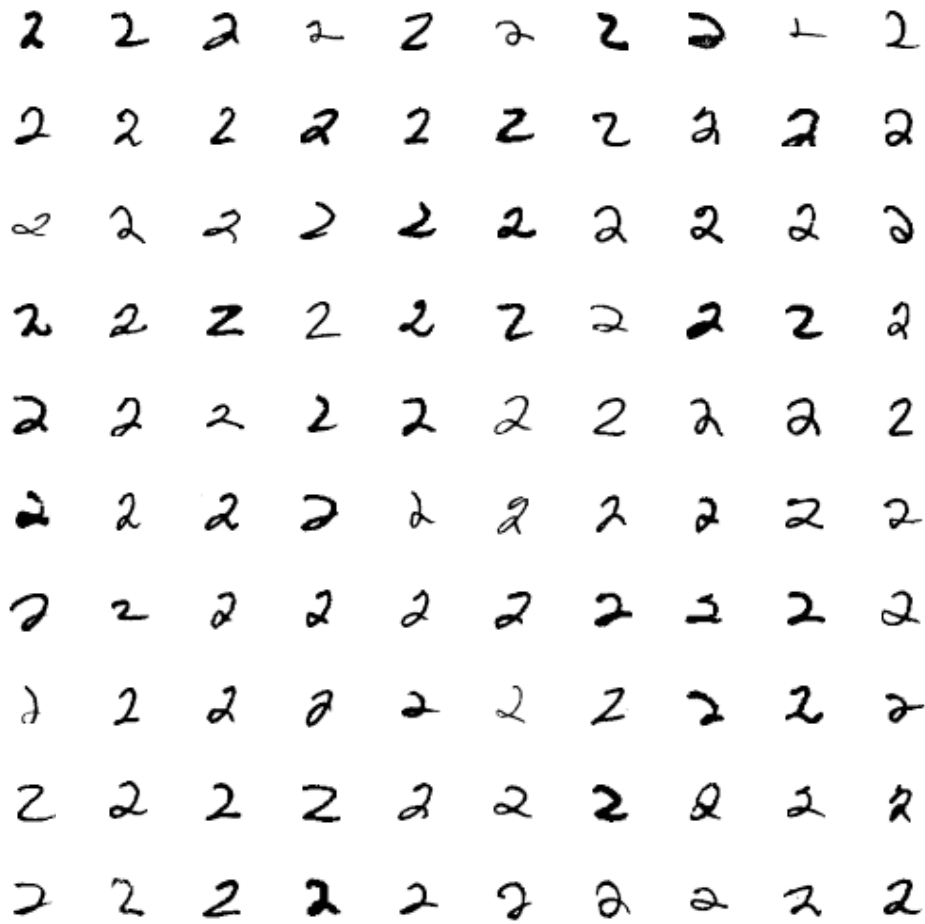

Figure 8: 100 digit "2" from training data.

