# Peer review of "On the Quantitative Analysis of Decoder-Based Generative Models"

_ICLR 2017 — accepted_

[Public Comment · (anonymous) · 05 Dec 2016]
**log-likelihood**

Very interesting paper to read. But I am a little confused that marginal log-likelihood (log p_\theta(x)) shown in this paper is usually more than 200, while the log p(x) shown in variational auto-encoder on mnist  is usually around -100(Both on 50 dimensions of VAE).  Does your paper show the same log p_\theta(x) as that in VAE?

Appreciate if you could solve my puzzle. Thanks.

[Official Review · AnonReviewer4 · rating 6 · confidence 5 · 14 Dec 2016]
**Application of BDMCMC to generative models. Relative contribution small.**

The paper describes a method to evaluate generative models such as VAE, GAN and GMMN. This is very much needed in our community where we still eyeball generated images to judge the quality of a model. However, the technical increment over the NIPS 16 paper: “Measuring the reliability of MCMC inference with bidirectional Monte Carlo” is very small, or nonexistent (but please correct me if I am wrong!).  (Grosse et al). The relative contribution of this paper is the application of this method to generative models. 
In section 2.3 the authors seem to make a mistake. They write E[p’(x)] <= p(x) but I think they mean: E[log p’(x)] <= log E[p’(x)] = log p(x). Also,  for what value of x? If p(x) is normalized it can’t be true for all values of x. Anyways, I think there are typos here and there and the equations could be more precise.
On page 5 top of the page it is said that the AIS procedure can be initialized with q(z|x) instead of p(z). However, it is unclear what value of x is then picked? Is it perhaps Ep(x)[q(z|x)] ?
I am confused with the use of the term overfitting (p8 bottom). Does a model A overfit relative to a another model B if the test accuracy of A is higher than that of B even though the gap between train and test accuracy is also higher for B than for A. I think not. Perhaps the last sentence on page 8 should say that VAE-50 underfits less than GMMN-50?
The experimental results are interesting in that it exposes the fact that GANs and GMMNs seem to have much lover test accuracy than VAE despite the fact that their samples look great.

[Official Review · AnonReviewer2 · rating 7 · confidence 4 · 16 Dec 2016]
**Great empirical work**

Summary:
This paper describes how to estimate log-likelihoods of currently popular decoder-based generative models using annealed importance sampling (AIS) and HMC. It validates the method using bidirectional Monte Carlo on the example of MNIST, and compares the performance of GANs and VAEs.


Review:
Although this seems like a fairly straight-forward application of AIS to me (correct me if I missed an important trick to make this work), I very much appreciate the educational value and empirical contributions of this paper. It should lead to clarity in debates around the density estimation performance of GANs, and should enable more people to use AIS.

Space permitting, it might be a good idea to try to expand the description of AIS. All the components of AIS are mentioned and a basic description of the algorithm is given, but the paper doesn’t explain well “why” the algorithm does what it does/why it works.

I was initially confused by the widely different numbers in Figure 2. On first glance my expectation was that this Figure is comparing GAN, GMMN and IWAE (because of the labeling at the bottom and because of the leading words in the caption’s descriptions). Perhaps mention in the caption that (a) and (b) use continuous MNIST and (c) uses discrete MNIST. “GMMN-50” should probably be “GMMN-10”.


Using reconstructions for evaluation of models may be a necessary but is not sufficient condition for a good model. Depending on the likelihood, a posterior sample might have very low density under the prior, for example. It would be great if the authors could point out and discuss the limitations of this test a bit more.


Minor:

Perhaps add a reference to MacKay’s density networks (MacKay, 1995) for decoder-based generative models.

In Section 2.2, the authors write “the prior over z can be drastically different than the true posterior p(z|x), especially in high dimension”. I think the flow of the paper could be improved here, especially for people less familiar with importance sampling/AIS. I don’t think the relevance of the posterior for importance sampling is clear at this point in the paper.

In Section 2.3 the authors claim that is often more “meaningful” to estimate p(x) in log-space because of underflow problems. “Meaningful” seems like the wrong word here. Perhaps revise to say that it’s more practical to estimate log p(x) because of underflow problems, or to say that it’s more meaningful to estimate log p(x) because of its connection to compression/surprise/entropy.

[Official Review · AnonReviewer3 · rating 7 · confidence 4 · 17 Dec 2016]
**Interesting work, their evaluation framework is available online, good contribution to the community**

# Review
This paper proposes a quantitative evaluation for decoder-based generative models that use Annealed Importance Sampling (AIS) to estimate log-likelihoods. Quantitative evaluations are indeed much needed since for some models, like Generative Adversarial Networks (GANs) and Generative Moment Matching Networks (GMMNs), qualitative evaluation of samples is still frequently used to assess their generative capability. Even though, there exist quantitative evaluations like Kernel Density Estimation (KDE), the authors show how AIS is more accurate than KDE and how it can be used to perform fine-grained comparison between generative models (GAN, GMMs and Variational Autoencoders (VAE)).

The authors report empirical results comparing two different decoder architectures that were both trained, on the continuous MNIST dataset, using the VAE, GAN and GMMN objectives. They also trained an Importance Weighted Autoencoder (IWAE) on binarized MNIST and show that, in this case, the IWAE bound underestimates the true log-likelihoods by at least 1 nat (which is significant for this dataset) according to the AIS evaluation of the same model.


# Pros
Their evaluation framework is public and is definitely a nice contribution to the community.

This paper gives some insights about how GAN behaves from log-likelihood perspective. The authors disconfirm the commonly proposed hypothesis that GAN are memorizing training data. The authors also observed that GANs miss important modes of the data distribution.


# Cons/Questions
It is not clear for me why sometimes the experiments were done using different number of examples (100, 1000, 10000) coming from different sources (trainset, validset, testset or simulation/generated by the model). For instance, in Table 2 why results were not reported using all 10,000 examples of the testing set?

It is not clear why in Figure 2c, AIS is slower than AIS+encoder? Is the number of intermediate distributions the same in both?

16 independent chains for AIS seems a bit low from what I saw in the literature (e.g. in [Salakhutdinov & Murray, 2008] or [Desjardins etal., 2011], they used 100 chains). Could it be that increasing the number of chains helps tighten the confidence interval reported in Table 2?

I would have like the authors to give their intuitions as to why GAN50 has a BDMC gap of 10 nats, i.e. 1 order of magnitude compared to the others?


# Minor comments
Table 1 is not referenced in the text and lacks description of what the different columns represent.
Figure 2(a), are the reported values represents the average log-likelihood of 100 (each or total?) training and validation examples of MNIST (as described in Section 5.3.2).
Figure 2(c), I'm guessing it is on binarized MNIST? Also, why are there fewer points for AIS compared to IWAE and AIS+encoder?
Are the BDMC gaps mentioned in Section 5.3.1 the same as the ones reported in Table2 ?
Typo in caption of Figure 3: "(c) GMMN-10" but actually showing GMMN-50 according to the graph title and subcaption.

[Public Comment · Martin Arjovsky · 29 Dec 2016]
**Comments**

I have mixed opinions about this work.

On the one hand I think it does a really nice job at providing a simple method to evaluate mode dropping and overfitting in generative models, which is one of the most important problems in machine learning now. I can see myself and many more people using these ideas to evaluate aspects of generative models

On the other hand, there is a constant use of the term "log-likelihood for GANs", which is extremely misleading. The log-likelihood of GANs is obviously ill-defined, and the assumption of "Gaussian observation" is incredibly false, since the true after-sampling variance that GANs are trained on is 0. You should relate how the model with noise added relates to properties on the actual distribution induced by the generator with no gaussian noise, instead of assuming the generator has Gaussian noise in the end, which is obviously false for a GAN and true for a VAE (which is trained this way since the reconstruction error assumes this kind of noise). I'm not asking for a precise mathematical statement of this relation, but I do think the writing sweeps this issue under the carpet, and the addition of isotropic noise on a distribution with an incredibly complex covariance matrix is more than questionable.

As a final note, I think section 2.2 which describes the central algorithm is very poorly written. After reading the section several times I have no idea how to implement it or how the sampling process is defined in the end. In my opinion, you should try to describe better:

- Which is the sampling distribution of the estimate of p(x) at time t. I believe this is p(x) = E_{z ~ p_t(z)} [ p(x | z) p(z) / p_t(z) ] but I'm far from sure.
- Explain why the successive estimator's are unbiased (if it's in the above form this is trivial)
- Explain why the estimators have less variance as t -> T.
- Explain what the theoretical guarantees of AIS are.

Minor:
- Typo in line 5 of page 4, "produces" -> "produce"
- I appreciate the wall clock times in figure 2.

Overall I think this is a cool paper, with some great consequences, but I think some issues (especially writing) should be polished.

[Author Response · Yuhuai Wu · 01 Jan 2017]
**A new version of the paper is updated**

A new version of the paper is updated. We revised the section 2.2 on the introduction to AIS, and several other places according to reviewers' comments.

[Public Comment · Matt Graham · 05 Feb 2017]
**Adaptive HMC implementation and Gaussian decoder model**

Thanks for the interesting paper.

I have a couple of queries / comments:

In the code you provide at

[Final Decision · Program Chairs · 06 Feb 2017]
**ICLR committee final decision**

This paper describes a method to estimate likelihood scores for a range of models defined by a decoder.
 
 This work has some issues. The paper mainly applies existing ideas. As discussed on openreview, the isotropic Gaussian noise model used to create a model with a likelihood is questionable, and it's unclear how useful likelihoods are when models are obviously wrong. However, the results, lead to some interesting conclusions, and on balance this is a good paper.